# Unraveling DPP4 Receptor Interactions with SARS-CoV-2 Variants and MERS-CoV: Insights into Pulmonary Disorders via Immunoinformatics and Molecular Dynamics

**DOI:** 10.3390/v15102056

**Published:** 2023-10-06

**Authors:** Arpan Narayan Roy, Aayatti Mallick Gupta, Deboshmita Banerjee, Jaydeb Chakrabarti, Pongali B. Raghavendra

**Affiliations:** 1National Institute of Biomedical Genomics, Kalyani 741251, West Bengal, India; anr19ms008@iiserkol.ac.in (A.N.R.); db3@nibmg.ac.in (D.B.); 2Department of Physics of Complex Systems, S. N. Bose National Centre for Basic Sciences, Block-JD, Sector-III, Salt Lake, Kolkata 700106, West Bengal, India; aayattigupta@gmail.com (A.M.G.); jaydeb@bose.res.in (J.C.); 3Technical Research Centre, S. N. Bose National Centre for Basic Sciences, Block-JD, Sector-III, Salt Lake, Kolkata 700106, West Bengal, India

**Keywords:** MERS-CoV, DPP4, SARS-CoV-2, RBD, spike protein variants, binding affinity

## Abstract

Human coronaviruses like MERS CoV are known to utilize dipeptidyl peptidase 4 (DPP4), apart from angiotensin-converting enzyme 2(ACE2) as a potential co-receptor for viral cell entry. DPP4, the ubiquitous membrane-bound aminopeptidase, is closely associated with elevation of disease severity in comorbidities. In SARS-CoV-2, there is inadequate evidence for combination of spike protein variants with DPP4, and underlying adversity in COVID-19. To elucidate this mechanistic basis, we have investigated interaction of spike protein variants with DPP4 through molecular docking and simulation studies. The possible binding interactions between the receptor binding domain (RBD) of different spike variants of SARS-CoV-2 and DPP4 have been compared with interactions observed in the experimentally determined structure of the complex of MERS-CoV with DPP4. Comparative binding affinity confers that Delta-CoV-2: DPP4 shows close proximity with MERS-CoV:DPP4, as depicted from accessible surface area, radius of gyration and number of hydrogen bonding in the interface. Mutations in the delta variant, L452R and T478K directly participate in DPP4 interaction, enhancing DPP4 binding. E484K in alpha and gamma variants of spike protein is also found to interact with DPP4. Hence, DPP4 interaction with spike protein becomes more suitable due to mutation, especially due to L452R, T478K and E484K. Furthermore, perturbation in the nearby residues Y495, Q474 and Y489 is evident due to L452R, T478K and E484K, respectively. Virulent strains of spike protein are more susceptible to DPP4 interaction and are prone to be victimized in patients due to comorbidities. Our results will aid the rational optimization of DPP4 as a potential therapeutic target to manage COVID-19 disease severity.

## 1. Introduction

DPP4 is a serine peptidase of highly conserved type II transmembrane glycoprotein, which comprises of a 6-residue N–terminal cytoplasmic tail, a 22-amino-acid transmembrane, and extracellular domain [1]. DPP4 is a multifunctional protein, also known as T-cell activation antigen CD26 [2], or adenosine deaminase binding protein (ADBP) [3] The extracellular domain cleaves dipeptide after the second position from the N-terminus of peptides with proline or alanine, indicating dipeptidyl peptidase activity [4]. DPP4 is widely distributed in the lung parenchyma, vascular endothelium, and fibroblasts of human bronchi, stating that DPP4 may play a vital role in modulating physiological and pathological functions in the lung [5]. Recently, DPP4 has obtained certain implications in the outbreak of the severe acute respiratory syndrome coronavirus 2 (SARS-CoV-2) infections, due to its ability as a cellular entry receptor or co-receptor for the virus [6,7,8].

It is interesting that several transmembrane proteins in the target cell apart from the primary receptor are invariably essential for the viral cell entry [9,10,11]. This is equally true for the coronavirus spike proteins, which can accept a widespread of cell-surface molecules along with the primary receptors for entry to the host. These additional attachment factors are referred to as co-receptors that are involved in vital roles in the dispersal of the virus [9,10,11]. Such virus infection responsiveness corresponds to co-receptor interactions which also triggers viral entry at a suitable place [9,10,11]. Studies depict that a receptor for a particular type of coronavirus can be employed for a cofactor of the infection of another or even a distant coronavirus. The human coronavirus OC43 (hCoV-OC43) receptor, O-acetylated sialic acid, supplements in the binding of hCoV-HKU1 to the cell surface [10,11]. HCoV-229E coronavirus uses the aminopeptidase N (APN) receptor [12]. In MERS-CoV, the carcinoembryonic antigen-related cell adhesion molecule 5 (CEACAM5) and glucose-regulated protein GRP78 are co-factors that strengthen the interaction in nonpermissive cells and entry in susceptible cells [9,11]. GRP78 can concurrently ingress the bat coronavirus HKU9 except for MERS-CoV [10]. It is also found CEACAM5 and CEACAM1 belong to the same family, which acts as the receptor of the animal coronavirus and mouse hepatitis virus [9,10,11]. Furthermore, in MERS-CoV, the tetraspanin CD9 act as a host cell-surface factor that enhances viral entry by scaffolding the receptor DPP4 and the protease TMPRSS2 [13]. TMPRSS2 plays an important role in the early infection of SARS-CoV, MERS-CoV, or SARS-CoV-2 [14]. Studies reveal that dendritic cell-specific inter-cellular adhesion molecule-3-grabbing nonintegrin (DCSIGN) and DCSIGN-related protein promotes SARS-CoV entry by interacting with spike protein [9,10,11]. DPP4 is a known receptor for MERS-CoV [15,16]. Coronaviruses use a variety of different receptors to enter host cells. This diversity of receptors makes it important to develop vaccines and therapies that target multiple receptors, in order to protect against a wide range of coronaviruses [12]. In SARS-CoV-2, DPP4 also plays a potential role as a co-receptor [6,17,18]. Bioinformatics based study on molecular docking approaches to predict the potential binding interactions between DPP4 and the RBD of SARS-CoV-2 is already there. However, the effects of DPP4 interaction due to spike protein mutations have not yet been explored. We are here interested in predicting the effect of SARS-CoV-2 variants related to DPP4 and their impact on comorbidities.

DPP4 is a notable receptor of MERS CoV [15]. The potential utilization of DPP4 as the binding target for MERS CoV is able to predict the specific potential molecular interactions of SARS CoV-2 with DPP4. There has been an immense focus in the reported literature on the ability of both SARS CoV-2 and SARS-CoV to bind to angiotensin-converting enzyme II (ACE2) protein to enter the host cells [14,19]. ACE2 is considered as the primary receptor for the spike protein of SARS CoV-2 to initiate infection [20]. Since the outbreak, many studies have been published describing the distribution of ACE2 receptor in the different types of human cells, such as lung, liver, kidney, and colon [20], which indicates that SARS CoV-2 may infect different organs in the human body. However, the primary target cells for SARS CoV-2 entry are the lung alveolar type 2 (AT2) cells, which express reduced levels of ACE2 [20], conferring the probable existence of co-membrane proteins facilitating host entry and infection. DPP4 interacts with several proteins that are important for viral processes and immune responses including ACE2, which implies a crosstalk between the two proteins that seek further inspection.

The most prevalent comorbidities in SARS CoV-2-infected patients are hypertension and diabetes, followed by cardiovascular and respiratory diseases [21]. Interestingly, DPP4 has a striking role in these disorders, especially in type 2 diabetes mellitus (T2DM). DPP4 plays a crucial role in glucose homeostasis via the proteolytic inactivation of peptides, such as glucagon-like peptide-1 (GLP-1), incretin hormones, and glucose-dependent insulinotropic polypeptide (GIP) [22]. DPP4 inhibitors are a class of medications that are used to treat type 2 diabetes. They work by blocking the action of dipeptidyl peptidase-4 (DPP4), an enzyme that breaks down certain hormones. These hormones, called incretins, help the body produce insulin and lower blood sugar levels. DPP4 concentration is higher in individuals with obesity than in those with normal body weight [23]. Barchetta et al. illustrates higher circulating DPP4 activity in non-alcoholic fatty liver disease (NAFLD) patients than in non-NAFLD patients [24]. The DPP4 inhibitor decreases plasma apolipoprotein B and triglyceride levels in patients with T2DM, indicating the function of DPP4 in regulating lipid metabolism [25]. DPP4 plays an important role in pulmonary impairment. Chronic obstructive pulmonary disease (COPD) related to impaired airflow imposes an inflammatory response by DPP4. DPP4 turns on CXCL12 which may further activate proteases either directly or via chemokine regulation to exacerbate tissue degradation in COPD [26]. DPP4 and a member of its gene family (DPP10) are also implicated in the pathophysiology of asthma [27,28]. Interleukin-13 (IL-13), secreted by Th2 cells, is related to airway inflammation and allergy in asthma [29]. DPP4 expression in pulmonary atrial smooth muscle cells mediates hypoxia-induced pulmonary hypertension [30,31]. DPP4 has identified roles in other infections. In chronic hepatitis C virus (HCV), DPP4 generates an antagonist form of the chemokine CXCL10 (also known as IP-10) by amino-terminal truncation of the protein [32], such that the elevation of plasma CXCL10 in patients with chronic HCV can modulate immune responses by chemokine receptor antagonism [33]. CXCR3 antagonism via truncated CXCL10 may also be an important regulatory mechanism occurring in tumors [34] and in sites of tuberculosis (TB) pathology [35]. Recently, there has been interest in using DPP4 inhibitors to treat COVID-19. The SARS-CoV-2 virus, which causes COVID-19, can bind to DPP4 to enter and infect cells. Therefore, DPP4 inhibitors could be used to block the virus from entering cells and preventing infection. Several clinical trials are currently underway to evaluate the safety and efficacy of DPP4 inhibitors in patients with COVID-19. Some of the early results of these trials have been promising, but more research is needed to confirm these findings. Overall, DPP4 inhibitors are a promising new class of medications that could be used to prevent and treat COVID-19. More research is needed to develop safe and effective DPP4 inhibitors that can be used to protect people from this deadly virus. Exploring the role of such a multi-faceted molecule in lung disease may yield vital and novel therapies.

Studies on the cell-surface co-factors facilitating the attachment and entry of SARS-CoV-2 are less known. There are few computational studies based on the interaction of the spike protein of SARS CoV-2 with DPP4 through molecular docking studies. This work is an extension of the work by Li et al. 2020 [6], where the potentiality of MERS-COV receptor DPP4 as a candidate binding target of the SARS-COV-2 spike was reported. However, the effects of DPP4 interaction due to spike protein mutations have not yet been explored. The varied emerging variants of SARS CoV-2 spike protein, from (i) alpha, (ii) beta, (iii) delta, (iv) gamma and (v) omicron (the list of mutations in each variant is given in Appendix A), are considered in the current study to investigate the binding interaction with DPP4. Such findings may highlight the role of DPP4 in SARS CoV-2 variants using the molecular docking and MD simulation approach. The results of our study will provide insight into the molecular mechanisms of viral entry and could have important implications for the development of new treatments for SARS-CoV-2. By comparing the predicted interactions between SARS-CoV-2 and DPP4 with those observed in MERS-CoV, our study seeks to clarify the potential role of DPP4 as a co-receptor for SARS-CoV2 and provide a basis for further investigation. We may analyse the effect of clustering of receptors like DPP4 and ACE2 both in association with SARS-CoV-2. From this we can predict the variant associated role in COVID as well as in long COVID. SARS-CoV-2 variants related DPP4 could influence COPD, TB, Cardio-pulmonary diseases, and comorbidities. Further studies are needed to fully understand the complex interaction between DPP4 and MERS-CoV, as well as to develop strategies to target DPP4 in the prevention and treatment of MERS-associated SARS CoV-2 or comorbidity-related diseases. The current study seeks to clarify the potential interaction of DPP4 with the different variants of SARS CoV-2 and compare it with MERS CoV through an MD simulation study. Using bioinformatics methods, this study identifies the high affinity between human DPP4 and the SARS-CoV-2 spike protein’s receptor-binding domain. Notably, this study is the first to report that the crucial binding residues of DPP4 are the same as those that bind to MERS-CoV-S.

## 2. Materials and Methods

### 2.1. Docking Studies

Docking between the spike protein of SARS CoV-2 in its native form (PDB id: 6M0J, chain B) and human DPP4 (PDB id: 4L72, chain A) has been performed in HADDOCK [36]. The docking protocol includes (i) rigid-body docking, (ii) semi-flexible refinement stage and (iii) final optimization in explicit solvent. In the initial rigid-body energy minimization stage, typically 1000 complex conformations are generated. The best 200 structures are selected for optimization through semi-flexible simulated annealing in torsion angle, followed by final short restrained molecular dynamic simulation in the explicit solvent. Clustering is performed using a cut-off of 7.5 Å and a minimum cluster size of 4. Thus, 200 structures are grouped in 10 clusters. The clusters are analysed and ranked according to their average interaction energies (sum of E_elec_, E_vdw_, and E_ACS_) and their average buried surface area. The resulting docked structures are sorted by minimum energy criteria and root mean square deviation (RMSD) clustering. The interface of the docked complex is analysed based on the Cα distance (5Å) between binding partner residues.

### 2.2. System Preparation

The crystal structure of MERS CoV spike protein complexed with human DPP4 (PDB id: 4L72) [15] is utilized for the all-atom MD simulations study. For Wild type (Wuhan)-spike: DPP4, we use the docked structure of the Wild type (Wuhan)-spike protein of SARS CoV-2 (PDB ID: 6M0J, chain B) [37], with that of DPP4 (PDB ID: 4L72, chain A). All the different Variants of Concern (VoC) that are now re-classified and updated as Variants being Monitored (VBM) of the spike protein of SARS CoV-2 bound with DPP4 (likewise, alpha-spike: DPP4; beta-spike: DPP4; delta-spike: DPP4; gamma-spike: DPP4 and omicron-spike: DPP4) are obtained by substituting the required amino acid alterations from the final structure of Wild type (Wuhan)-spike: DPP4 at a 500 ns time span. The list of mutations for each variant is tabulated in detail in Appendix A. Thus, we have considered a total of seven different systems for MD simulation studies.

### 2.3. MD Simulation Studies

We perform molecular dynamics (MD) simulations of all the systems in explicit water. The Spc216 (Simple Point Charge) water molecule [38] is used for solvation under a dodecahedron box with a minimum distance between the protein and the box 1.0 Å. Counter-ions (Na^+^ and Cl^−^) are added to make the system electro-neutral. The GROMACS 2018.6 program with GROMOS96 53a6 force field [39] has been applied at 300 K and 1 atm pressure in an isothermal–isobaric ensemble using periodic boundary conditions and 2 femtosecond time-step. The longer-ranged coulombic interactions are treated using the particle-mesh Ewald method [40]. The total number of particles is maintained to be the same in all simulations (N = 164,301) to make the simulated ensembles equivalent. Simulation has been performed using NPT, where the number of particles, pressure and temperature are kept constant. The simulation trajectories are calculated up to 500 ns. The equilibrations of the simulated structures are assessed from the saturation of the root mean squared deviations (RMSD). Various analyses are carried out from the converged trajectory with tools in GROMACS to examine the system properties. The trajectories are being loaded for calculation and visualization using VMD [41].

### 2.4. Protein Stability and Flexibility Analysis

The DYNAMUT tool [42] has been used to analyse the effect of all mutations present on the RBD domain and outside the RBD (truncated S1 domain) of the spike protein; free energy changes (ΔΔG) and vibrational entropy difference (ΔΔS) were calculated. Protein flexibility and stability were examined on the complexes based on these calculations by utilizing the normal mode analysis (NMA)-based elastic network contact model (ENCoM). ENCoM is an Elastic Network Contact Model that applies a potential energy function and includes a pairwise atom-type non-bonded interaction term to add an extra layer of information regarding the effect of the specific nature of amino acids on dynamics within the context of NMA. ENCoM tries to approximate ΔΔG through the calculations of the vibrational entropy (Δ*S*) of wild-type and mutant structures.

Δ*S* between two conformations (*A*, *B*) in terms of their respective sets of eigenvalues is given by $∆S_{Vib,A\to B}=h\times ln\left(\frac{\prod_{n=7}^{3N}\lambda_{n,A}}{\prod_{n=7}^{3N}\lambda_{n,B}}\right)$.

*h* represents the Boltzmann constant, where $\lambda_{n,i}$ represents the *n*th normal mode.

### 2.5. Perturbation Residue Scanning (PRS)

PRS has been performed using the pPerturb server [43,44]. It allows the mutation of one or more residues to alanine and generates a perturbation profile (ΔQ vs. C_alpha_-C_alpha_) distance from the perturb site. The ΔQ value is the magnitude of perturbation experienced by each residue and C_α_-C_α_ is the distance from the perturbed site. The perturbation effect can be analysed as a distance connecting the perturb residue to its nearby residues or on the interaction network strength. “Coupling distance” (d_c_) refers to a measure of the degree of coupling between two residues in a protein structure. This measurement is based on the interaction energy between residues and can provide insight into how changes in a protein’s structure can affect its function. The coupling distance can be used to predict the stability and functionality of a protein.

## 3. Results

To predict the specific binding potential of SARS CoV-2 with DPP4, a molecular docking approach with HADDOCK has been followed. HADDOCK uses a combination of experimental data and physical principles to generate models that are both accurate and informative. It has been used to predict the structure of the SARS-CoV-2 spike protein in complex with its receptor, ACE2 [45], to study the interaction between the PD-1 and PD-L1 proteins [46], and to study the interaction between the HIV-1 gp120 protein and its receptor, CD4 [47]. This information has been used to develop new vaccines and therapies for COVID-19, new cancer immunotherapy drugs, and new antiretroviral drugs. In short, HADDOCK is a powerful tool for understanding and manipulating protein–protein interactions, which can be used to develop new drugs and therapies for a wide range of diseases. The native form of the SARS CoV-2 spike protein is in its open conformation. It is a ligand binding conformation. We have considered only the RBD region instead of the entire trimeric spike. The overall orientation of the predicted binding interactions bears similarity to the predictions of Li et al., 2020 [6] and Cameron et al., 2021 [8] but do not reproduce their findings completely. A number of new, additional binding interactions are also noticed at the binding interface. The docking poses with the lowest interaction energy are further selected for MD simulation study.

A 500 ns long MD simulation is carried out on DPP4 and spike protein variants from MERS CoV and SARS CoV-2 complexes. The equilibrations of the simulated structures are assessed from the RMSD. RMSD is a quantitative measure of the variation of protein complex considering all the non-hydrogen atoms concerning the initial conformation of protein complex along the time of simulation, shown in the Appendix A. The final structure of each of the systems given by a snapshot is also shown in (Figure 1). We observe that all the systems equilibrate within 500 ns of simulation time. The binding interactions between the RBD of SARS CoV-2 variants and DPP4 are compared to identify the most similar interactions to those of the MERS CoV: DPP4 complex. The changes in the binding state of different systems are manifested by several parameters, detailed below.

### 3.1. Comparative Analysis of Spike Variants and DPP4 Interaction

#### 3.1.1. Secondary Structure

We have used DSSP to assign the secondary structural elements to the residues. The secondary structural elements play a vital role in mediating interactions in the binding interface. Studies on the structural analysis and comparison of protein–protein interfaces depict regular secondary structures (helices and strands), which are the main components of the protein homodimer (obligate interface), whereas non-regular structures (turns, loops, etc.) frequently mediate interactions in the heterodimeric protein–protein interfaces [48]. Secondary structural changes of DPP4 and spike protein for each of the ensembles from the converged trajectory have been studied. We find that the secondary structure of DPP4 remains the same in all the complexes during various time frames of MD simulation (Figure 2). The spike protein, however, undergoes substantial changes in secondary structure across the time span of simulation (Figure 3). In delta SARS CoV-2:DPP4, the spike protein, a non-regular secondary structural element, is in abundance in comparison to the rest. In comparison to MERS CoV: DPP4, the turn is found to decrease to a certain extent in the case of SARS CoV-2 variants. The spike protein of MERS CoV is less α-helical than that in the variants of SARS CoV-2. The prevalence of non-regular secondary structural elements in MERS CoV and the delta SARS CoV-2 RBD region account for a potent binding partner to DPP4 to form the heterocomplex. The frequency of non-regular secondary structural elements (turns, loops, etc.) plays vital role to mediate the heterodimeric protein–protein interaction interface [48]. Thus, this can be justified by their predominance in MERS CoV and the delta SARS CoV-2 RBD region for being a better binding partner of DPP4.

#### 3.1.2. Radius of Gyration

We characterize the comparative binding affinity of the complex through its radius of gyration (Rg). Rg of a complex measures its compactness. It is calculated as the average distance of the C-alpha atoms from their centre of mass. Rg is computed for every ensemble and the mean value is shown in Table 1. MERS CoV: DPP4 shows the minimum mean value followed by delta SARS CoV2:DPP4 and gamma SARS CoV2:DPP4. Histogram distribution is generated for each system separately (Figure 4A). It is found that all the histogram distributions are symmetric and unimodal. The peak height of the histogram is observed lower in the case of MERS CoV: DPP4 and gamma SARS CoV2: DPP4. A sharp peak in delta SARS CoV2:DPP4 with a lower Rg score impacts more compactness. An increase in Rg values in the rest of the system implies a decrease in protein compactness due to DPP4 interaction. However, when DPP4 binds with the spike protein of MERS CoV, delta SARS CoV-2 and gamma SARS CoV-2, there is a conformational change that alters the Rg value. Lower Rg value in them imparts a more tight binding in comparison to the rest of the ensemble. 

#### 3.1.3. Accessible Surface Area

The contact surface at the binding interface depicted by the solvent-accessible surface area (SASA) is used as a descriptor to measure the relative DPP4 binding strength to various spike proteins. The less the surface area of a biomolecule in a complex that is accessible to a solvent, the larger the binding interface. The mean value of SASA for each of the studied systems is given in Table 1. Less accessible surface area for the delta SARS CoV-2:DPP4 and MERS CoV: DPP4 complex confers stronger binding than that of the other interacting systems. The probability distribution of SASA, H(SASA) is shown in Figure 4B. The bell-shaped curve can be visualized for the different ensembles. The distribution of Wild type (Wuhan)-spike SARS CoV-2:DPP4 and beta SARS CoV-2:DPP4 is found almost uniform. An increase in accessible surface area in alpha SARS CoV-2:DPP4 and omicron SARS CoV-2:DPP4 decreases the binding affinity at the protein–protein interfaces. In delta SARS CoV-2:DPP4, the histogram distribution is slightly left-skewed, with a lower value of SASA similar to MERS CoV: DPP4, indicating that such an interaction buries a large accessible surface area which explains their strong binding.

#### 3.1.4. Hydrogen Bonds

The hydrogen bonds formed between DPP4 and spike protein variants of SARS CoV-2 and MARS CoV are calculated for all the ensembles. Such an analysis gives the number of hydrogen bonds at a distance of less than 3.5 Å between all possible donors D and acceptors A with a D-H-A angle of 180° to 30°. The mean value of the number of hydrogen bonds (n_H_) due to DPP4-spike interactions for each system is tabulated in Table 1. Overall, the number of hydrogen bonds is quite high in delta SARS CoV-2:DPP4 followed by MERS CoV: DPP4. The distribution of n_H_, H(n_H_) is unimodal. An increase in the number of hydrogen bonds between delta SARS CoV-2 and DPP4 denotes stronger interaction between them. A similar observation is found in MERS CoV: DPP4 (Figure 4C). The Favourable number of intermolecular hydrogen bonds involves high-affinity binding in such systems.

#### 3.1.5. Interface Analysis

We have also analysed the closeness between the two binding partner-DPP4 and spike from each of the ensembles. The binding interface has been studied by calculating the centre of mass of DPP4 and spike as a function of time (Figure 5). The protein–protein interface within a 0.5 nm distance in each of the complexes is tabulated (Table 2). The key residues forming the binding partner between MERS CoV and DPP4 as characterized by the X-ray diffraction data of their complex remain conserved till the end of the simulation (Table 2). Since SARS CoV-2 and MERS CoV are related viruses, an important assumption in these studies is that the RBD of SARS CoV-2 would likely bind to DPP4 as MERS CoV with a similar conformation.

Interface analysis confirms that delta SARS CoV-2:DPP4 has a greater number of interaction than in other systems. The binding interface and the interacting partners with the mode of the interactions are tabulated for each system (Appendix A). We also show the frequency (%) of different interactions in Appendix A. Some illustrative cases of common interfacial interactions are shown in Appendix A. The interface is characterized by various mode of interaction. This is to mention that the number of hydrogen bond at the interface is not particularly high for any of the complexes. As the mutations involve a large number of charged residues, the formation of salt bridges has been seen at the binding interface. Our study has identified that A282 and A289 of DPP4 are found to interact with the V483 and Q493 of spike variants, respectively, in all the cases. Thus, it forms the common interface. It is important to note that spike protein residues showing a mutation in the delta variant, L452R and T478K, directly participate in DPP4 interaction. Thus, the mutation in the delta variant enhances the binding. E484K in alpha and gamma variants of spike protein is also found to interact with DPP4. Thus, DPP4 interaction with spike protein becomes more suitable due to L452R, T478K and E484K mutation. The more virulent strains of spike protein are more susceptible to DPP4 interaction and hence are prone to be victimized in patients due to comorbidities.

### 3.2. Fluctuations in the Complexes

#### 3.2.1. RMSF

The intrinsic dynamics of the binding site have been analysed in our current study. Flexible and rigid regions of DPP4 and spike variants from each of the complexes are compared to root mean square fluctuations (RMSF) analysis. RMSF considers the overall magnitude of the fluctuation of each Cα-atom. We have calculated RMSF separately for DPP4 and spike variants. RMSF for DPP4 in all the complexes is almost equal (Figure 6A). Overall fluctuation in DPP4 protein is not so profound. In the case of spike protein variants, we find perturbations due to delta SARS CoV-2 and alpha SARS CoV-2. Such a high degree of flexibility in the RBD region of the delta variant is likely to be necessary for it to be able to interact with DPP4 (Figure 6B). Flexible regions in globular protein mediate protein–protein interactions [49]. Moreover, the involvement of the flexible loop region at the binding interface imposes stronger interaction [50]. In the current study, such high degree of flexibility in the RBD region of the delta variant is likely to be necessary for it to interact with DPP4 because flexibility is related with less stability. In order to gain stability, such a flexible and fluctuating candidate acts as a potent binding partner of DPP4. Protein–protein interaction through complex formation helps it attain its stability and hence its flexibility decreases.

#### 3.2.2. PCA

Principal component analysis (PCA) utilizes a covariance matrix that includes information on the anisotropy of such fluctuations and their correlations between sites. PCA in MD simulation identifies only the essential motion eliminating other rotational and translational movements. PCA analysis for DPP4 is similar in all the ensembles (Figure 6C). However, the PCA of spike protein variants shows remarkable variations. In the spike protein of beta SARS CoV-2, fluctuation is found prominent (Figure 6D).

#### 3.2.3. Protein Stability

From the MD simulation study, we can observe that spike protein residues having mutations L452R and T478K in the Delta variant, and E484K in the Gamma and Alpha variants are found to directly participate in DPP4 interaction. Hence, we have chosen such an interacting interface to analyse the effect of mutations on protein structures by calculating the change in vibrational entropy (ΔΔS) and change in folding free energy (ΔΔG) as a thermodynamic state function. We have applied normal mode analysis (NMA) utilizing harmonic motions in a system to provide insights into its dynamics and accessible conformations due to such mutations in the spike protein variants. NMA is a valuable tool for studying protein dynamics and interactions. It is performed by calculating the eigenvalues and eigenvectors of the Hessian matrix of the molecule. The natural frequencies and vibrational modes of the molecule represent the different ways in which it can move [51,52]. NMA has been used to study a variety of protein dynamics and interactions, including the effects of mutations on protein flexibility and stability. The entropy and folding free energy changes are indicated in (Appendix A). In the Delta variant, L452R and T478K mutations are favoured by the attainment of molecular stability (0.692 Kcal/mol and 0.877 Kcal/mol, respectively), and vibrational entropy changes (−0.154 kcal mol^−1^ K^−1^ and −0.728 kcal mol^−1^ K^−1^, respectively) leading to a decrease in molecular flexibility. It is found that due to the L452R mutation conventional hydrogen bond between V350 and L452 breaks, and the hydrophobic bond between L452 and L492 also breaks. In the case of T478K, intra-molecular interaction remains the same before and after mutation. On the other hand, E484K mutation for Gamma and Alpha variants elicits a destabilizing effect (−0.238 kcal/mol) on the protein and the vibrational entropy energy values (0.453 kcal mol^−1^ K^−1^) attributed towards the increase in molecule flexibility. It appears that the E484K mutation causes the formation of one carbon–hydrogen bond between K484 and C488.

#### 3.2.4. Perturbation Residue Scanning (PRS)

Additionally, to probe the effect of the interface mutations on the allosteric residue, PRS has been analysed. PRS is used for predicting the energetic couplings and stability changes in proteins upon mutations. It is based on a perturbation approach that has been validated against a large dataset of experimental data. Such a calculation was widely used in a number of previous studies to obtain reliable predictions of protein stability and energetic couplings. This has been used to predict the effects of mutations on the stability of enzymes, the binding of proteins to ligands, and the allosteric regulation of proteins. The perturbation effect can be analysed as a distance connecting the perturb residue to its nearby residues or on the interaction network strength. The PRS analysis of the most relevant interface mutations of the RBD of Delta (L452R, T478K), Gamma (E484K) and Alpha (E484K) has been conducted. In the case of the Delta variant, perturbation analysis of R452 results in a coupling distance of 3.3 ± 0.2 Å and K478 results in a coupling distance (d_c_) of 4.2 ± 0.4 Å, and perturbation analysis of K484 (in case of the Gamma and Alpha variants) reveals a d_c_ of 2.5 ± 0.1 Å (Appendix A). If a mutation occurs at a residue with a high coupling distance, it is less likely to affect the stability of its interacting partner residues compared to a mutation at a residue with a low coupling distance. Therefore, in the Delta variant, the L452R mutation has the maximum effect on the adjacent residues. We have also shown the residue-wise Perturbation Residue Scanning profile (ΔQ) distance from the perturb site (Appendix A) as the magnitude of perturbation as one of the factors in determining the effect of a mutation on adjacent residues. The coupling distance is uniquely sensitive to the environment of a residue in the protein to a distance of ∼15 Å [53]. We have also found adjoining residues which are causing maximum perturbation with a cut-off distance of 15Å due to mutation L452R, T478K and E484K (Appendix A). The maximum perturbation is observed in Y495 due to the L452R mutation. Similarly, Q474 and Y489 show significant perturbation due to the T478K and E484K, respectively. Other factors, such as the location of the mutated residues within the protein structure and their interactions with other residues, can also affect the extent of perturbation and its effect on adjacent residues.

## 4. Discussion

The COVID-19 pandemic has become more dangerous as new SARS-CoV-2 variants have appeared in each new wave of infection. The primary functional receptor for SARS-CoV-2 is ACE2, DPP4, has also been suggested to be its potential co-receptor for viral cell entry [54,55]. To predict the specific potential interaction between DPP4 and various spike variants from SARS CoV-2, we have employed molecular docking and MD simulation studies. Notably, the interaction between MERS-CoV spike protein and DPP4 is essential for viral infection and is correlated with susceptibility to MERS-CoV infection, as well as with viral genome detection in the culture medium of infected cells [13]. The binding interactions between MERS CoV and DPP4 have been characterized by the X-ray crystal structure of their complex. Such an analysis of the binding interactions is very useful for comparison with our MD simulation study. Interface analysis has shown that delta SARS CoV-2:DPP4 has a greater number of interactions than in other systems. Mutations can have varying effects on protein–protein interactions. Depending on the specific location and nature of the mutation, it can either strengthen or weaken the interaction between two proteins. In some cases, a mutation may create a new interaction site on one of the proteins or alter an existing interaction site, leading to stronger binding between the two proteins. This can result in enhanced biological activity or stability of the complex formed by the two proteins. The majority of these mutations have accumulated primarily in the spike glycoprotein. Since the Spike protein serves as the main infection target, focusing on spike protein mutations increases transmission effectiveness and allows for resistance to neutralizing antibodies. Mutations on the RBD region of the spike have been reported to increase affinity to the ACE2 receptor.

The current study elicits that spike protein residues showing mutations like L452R and T478K in the Delta variant, and E484K in the Gamma and Alpha variants directly engages in DPP4 interaction. Bioinformatics techniques based on the protein crystal structures have speculated that the MERS-CoV receptor DDP4 might act as a candidate-binding target of SARS-CoV-2 as well [6,7,8]. It is found that the key interacting sites in DPP4 are similar with those bound to the MERS-CoV spike RBD. The overall orientation of the predicted binding interactions bears uniformity to the predicted literature. Our study supports the previous findings and further detects the mutations at E484, L452R and T478K of different spike variants to be crucial to the acquirement of this binding ability. Binding affinity for each of the systems has been compared with the interactions between MERS CoV and DPP4. Interpretation of DPP4-spike variant binding with SARS CoV-2 using molecular simulation-based recognition can be an effective approach to determine binding affinity considering the radius of gyration, solvent-accessible surface area, hydrogen bonding, and total energy of interaction. Such a study denotes that MERS CoV and delta SARS CoV-2 have a stronger binding affinity for DPP4. The distance between the centre of masses between the two proteins infers the interacting interface. The overall interaction is well built in the spike protein of the delta variant with DPP4, as known from the interacting interface. Further, the effect of mutations has been predicted on the protein stability and flexibility through the vibrational entropy change (ΔΔS) and free energy change (ΔΔG) calculations. In the Delta variant, L452R and T478K both have a stabilizing effect on DPP4 interactions. Loss of flexibility due to such mutations in Delta-SARS CoV-2: DPP4 concludes a better binding interaction between the two proteins.

Overall, our study compared DPP4 interaction with MERS, Wildtype and all five variants of SARS-CoV-2, and identified unique RBD residues crucial for the interaction with the DPP4 receptor. By comparing the predicted interactions between all SARS-CoV-2 variants and DPP4 with those observed in MERS-CoV, such a study seeks to clarify the potential role of DPP4 as a co-receptor for SARS-CoV-2 and provide a basis for further investigation. In a study to predict human proteins likely to interact with SARS-CoV-2 spike protein based on the available crystal structures by interolog mapping, domain–domain interaction, and machine learning approaches [56,57,58,59], ACE2 and DPP4 are the top-ranking host factors, as ACE2 is the known receptor of SARS-CoV-2. So, there is rising interest in DPP4 which acts as an attachment factor in MERS-CoV. The immunological characterization analysis elicits that the infection of SARS-CoV-2 in immune cells is not related with the expression of ACE2 [17]. Thus, it seems that ACE2 should have utilized some helpers in the swift infection and pathogenesis of SARS-CoV-2. Following the study of Vankadari and Wilce, 2020 [7], Li et al., 2020 [6], and Cameron et al., 2021 [8], we tried to focus on the potential role of DPP4 as a co-receptor for SARS-CoV-2 variants. Virulent strains of spike protein are more susceptible to DPP4 interaction and are prone to be victimized in patients due to comorbidities. Our results will aid in the rational optimization of DPP4 as a potential therapeutic target to manage COVID-19 disease severity. By analysing the clustering of receptors like DPP4 and ACE2 in association with SARS-CoV-2, we can hypothesize the role of variants in COVID-19 and long COVID-19 due to DPP4 binding. This claim is supported by the observation that survivors of previous coronavirus infections, such as SARS and MERS, have shown a similar constellation of persistent symptoms [60,61,62], raising concerns about clinically significant sequelae of COVID-19. Understanding the interface and the nature of interaction between spike variants and DPP4 is essential for predicting the binding affinity of all future emerging variants. DPP4 circulates in the plasma and has a multidimensional role in immune regulation, inflammation, oxidative stress, cell adhesion and apoptosis by targeting different substrates. DPP4-mediated comorbidities due to SARS CoV-2 infections among patients with diabetes, hypertension, and cardiovascular diseases are more common. The more virulent strains of spike protein are more susceptible to DPP4 interaction and hence are prone to be victimized in patients due to the comorbidities.

## Figures and Tables

**Figure 1 viruses-15-02056-f001:**
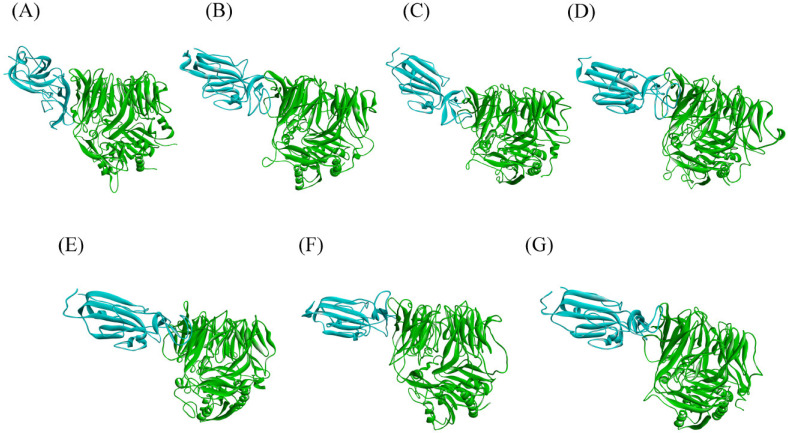
Snapshot of DPP4 (green) and spike protein (cyan) complex of (**A**) MERS CoV, (**B**) Wild type (Wuhan)-spike: DPP4, (**C**) alpha-spike: DPP4, (**D**) delta-spike: DPP4, (**E**) gamma-spike: DPP4, (**F**) omicron-spike: DPP4, (**G**) and beta-spike: DPP4 at 500 ns time span.

**Figure 2 viruses-15-02056-f002:**
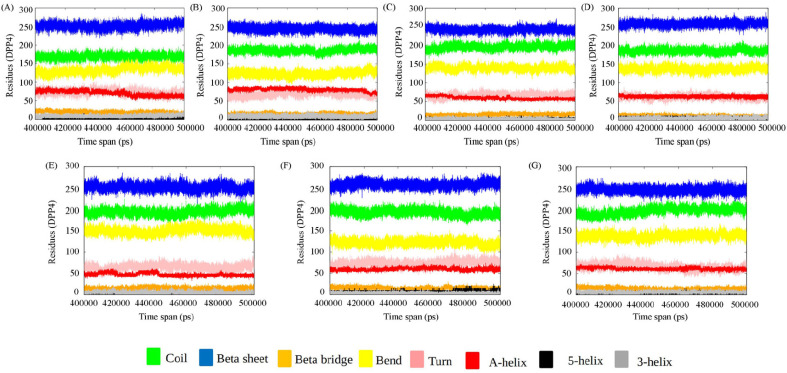
Secondary structure of DPP4 of the complexes during various time frame of MD simulation. (**A**) MERS CoV:DPP4, (**B**) wild-type SARS CoV-2:DPP4, (**C**) Alpha SARS CoV-2:DPP4, (**D**) Beta SARS CoV-2:DPP4, (**E**) Delta SARS CoV-2:DPP4, (**F**) Gamma- SARS CoV-2:DPP4, and (**G**) Omicron SARS CoV-2:DPP4.

**Figure 3 viruses-15-02056-f003:**
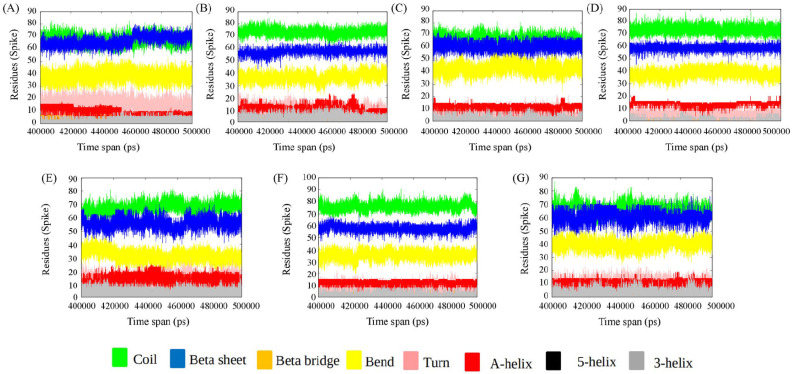
Secondary structure analysis of spike protein of (**A**) MERS CoV:DPP4, (**B**) wild-type SARS CoV-2:DPP4, (**C**) Alpha SARS CoV-2:DPP4, (**D**) Beta SARS CoV-2:DPP4, (**E**) Delta SARS CoV-2:DPP4, (**F**) Gamma- SARS CoV-2:DPP4, and (**G**) Omicron SARS CoV-2:DPP4.

**Figure 4 viruses-15-02056-f004:**
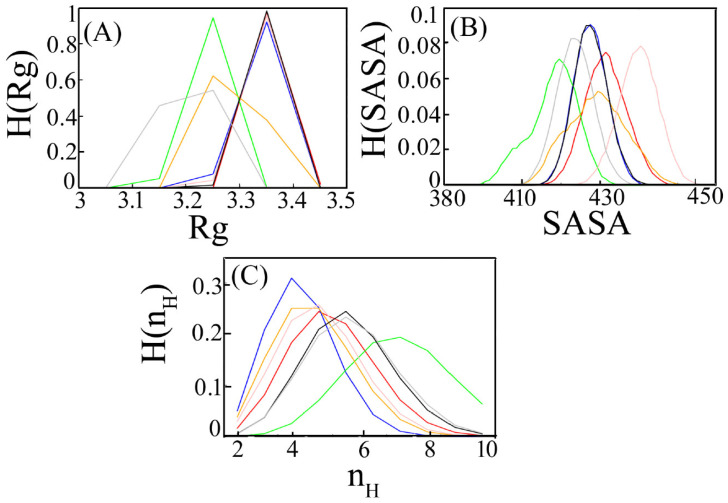
Histogram distribution from converged trajectory of (**A**) radius of gyration (Rg), (**B**) solvent-accessible surface area (SASA) and (**C**) hydrogen bond number (n_H_). The colour demarcation of different systems is as MERS-CoV:DPP4 is marked in grey, wild-type SARS CoV:DPP4 in black, alpha SARS CoV:DPP4 in red, beta SARS CoV:DPP4 in blue, delta SARS CoV:DPP4 in green, gamma SARS CoV:DPP4 in orange and omicron SARS CoV:DPP4 in pink.

**Figure 5 viruses-15-02056-f005:**
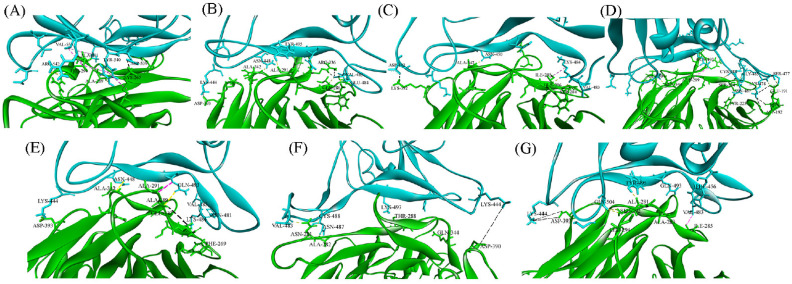
Representation of interface interacting residues involved in hydrogen, hydrophobic and salt bridge bond interaction in the representative structure of (**A**) MERS CoV (**B**) wt-spike: DPP4 (**C**) alpha-spike: DPP4 (**D**) delta-spike: DPP4 (**E**) gamma-spike: DPP4 (**F**) omicron spike: DPP4 and (**G**) beta-spike: DPP4 complex obtained after clustering analysis. Spike variants and DPP4 are represented as cartoon structures and surface diagrams with colour cyan and green, respectively. Hydrogen, hydrophobic and salt bridge bonds are shown in yellow, pink and black dashed lines, respectively.

**Figure 6 viruses-15-02056-f006:**
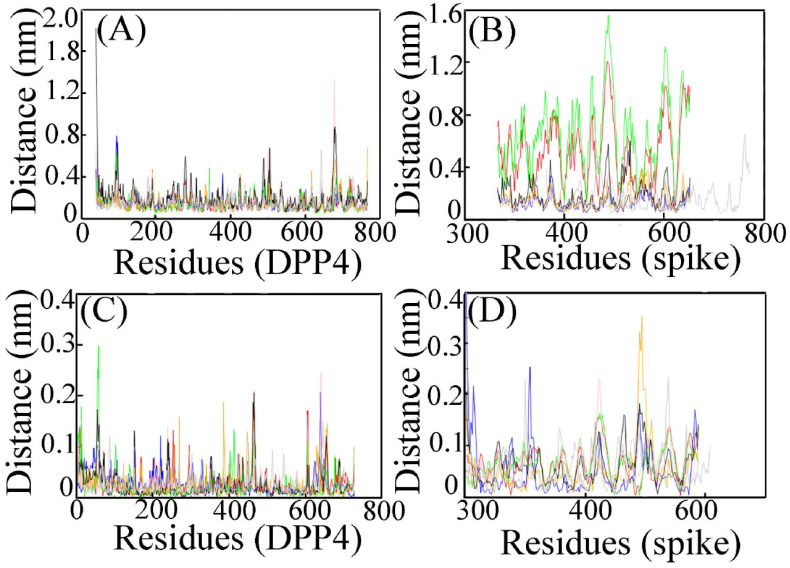
(**A**) RMSF analysis for DPP4, and (**B**) RMSF of spike protein of MERS CoV and SARS CoV-2 variants. (**C**) PCA analysis for DPP4 of the complexes. (**D**) PCA analysis for spike variants of the complexes. Colour demarcations of the systems are same as Figure 4.

**Table 1 viruses-15-02056-t001:** The calculated mean value of radius of gyration, solvent assessable surface area and number of intermolecular hydrogen bonds for each system.

System	Rg (nm)	SASA (nm/S2/N)	nH (no.)
MERS CoV:DPP4	3.200	4.135	4.208
WT-spike SARS CoV-2:DPP4	3.340	4.176	4.102
Alpha SARS CoV-2:DPP4	3.358	4.214	3.494
Beta SARS CoV-2:DPP4	3.328	4.175	2.375
Delta SARS CoV-2:DPP4	3.239	4.079	5.998
Gamma SARS CoV-2:DPP4	3.289	4.191	2.832
Omicron SARS CoV-2:DPP4	3.344	4.299	3.057

**Table 2 viruses-15-02056-t002:** The binding interface at a distance of 0.5 nm between spike protein of SARS CoV-2 variants, 320 MERS and DPP4.

Delta		Wildtype		Gamma		Alpha		Beta		Omicron		MERS	
DPP4 Residue	SARS-CoV-2 Residue	DPP4 Residue	SARS-CoV-2 Residue	DPP4 Residue	SARS-CoV-2 Residue	DPP4 Residue	SARS-CoV-2 Residue	DPP4 Residue	SARS-CoV-2 Residue	DPP4 Residue	SARS-CoV-2 Residue	DPP4 Residue	MERS Residue
A282 A289 A291 K287 D331 K392 E191 R336 K190 E347 E332 D390 D192 R343 S284 S284 S284 Q344 Y225 E191	V483 Q493 Q493 T478 R452 D442 K478 E471 E484 K444 R452 K444 K478 D442 C488 N487 G485 G504 N487 S477	A282 A289 A291 F269 I285 A291 A342 Q286 D331 R336 K392 D393 R343	V483 Q493 Q493 V483 V483 Y495 N448 E484 S494 E484 D442 K444 D442	A282 A289 A291 F269 I285 I287 Q286 Q444 E332 D393 A342	V483 Q493 Q493 V483 V483 V483 N481 K444 K484 K444 N448	A282 A289 F269 I485 I287 Q344 K392 R336 S284 Q286 A342	V483 Q493 V483 V483 V483 K444 D442 E471 K484 K484 N450	A289 A291 A291 I285 E347 E378 D393 A291 L294 S292	Q493 F456 Q493 V483 K444 K444 K444 Y495 G502 G504	A282 A289 Q344 T288 D390 N281 A282	V483 Q493 K444 Y505 K444 C488 N487	K267 R317 R336 L294 A289 A291 L294 L294	D539 D510 Y499 R542 K502 L506 Y540 V555

## Data Availability

Not applicable.

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
