# Peer review of "Unraveling DPP4 Receptor Interactions with SARS-CoV-2 Variants and MERS-CoV: Insights into Pulmonary Disorders via Immunoinformatics and Molecular Dynamics"

_viruses, 2023, doi:10.3390/v15102056_

Round 1

Reviewer 1 Report

Journal: viruses

Manuscript ID: 2629526

Title:  Unraveling the Interactions between Human DPP4 Receptor, SARS-CoV-2 Variants, and MERS-CoV converged for Pulmonary Disorders Integrating through Immunoinformatics and Molecular Dynamics

Authors investigated the interactions of spike protein variants with dipeptidyl peptidase 4 (DPP4 also known as (CD26)) through molecular docking and simulation studies.

The possible binding interactions between receptor binding domain (RBD) of different spike variants of SARS-CoV-2 and DPP4 are compared with interactions observed in experimentally determined structure of complex of MERS-CoV with DPP4. Comparative binding affinity suggests that Delta-CoV-2: DPP4 shows close proximity with MERS-CoV:DPP4, according to the analysis of accessible surface area, radius of gyration and number of hydrogen bonding in the interface.

Authors compared DPP4 interaction with MERS, wild type (Wuhan), and five variants of SARS-CoV-2 and identified unique RBD residues crucial for the interaction with the DPP4 receptor.

It is claimed that DPP4 may be a potential therapeutic target to manage COVID-19 disease severity.

MAJOR

1. This work is an extension of the work:

Reference 6

Li, Y. et al. The MERS-COV receptor DPP4 as a candidate binding target of the SARS-COV-2 spike. iScience 23, 503 101160 (2020).

In this work the high affinity between human DPP4 and the spike (S) receptor-binding domain of

SARS-CoV-2 was found using bioinformatics methodologies. They were first in noticing that the crucial binding residues of DPP4 are identical to those that are bound to the MERS-CoV-S.

2. Variants of concern of SARS-CoV-2 are analysed: Alpha, Beta, Delta, Gamma, and Omicron.

An increase in accessible surface area in Alpha SARS-CoV-2: DPP4 and Omicron SARS CoV-2: DPP4 decreases the binding affinity at the protein-protein interfaces. Currently, the Delta variant is no longer of concern.

 3. Besides DPP4 there are other receptors employed by mammalian coronaviruses: the murine carcinoembryonic antigen-related cell adhesion molecule 1a (mCEACAM 1a), aminopeptidase N (ANPEP also CD13), and angiotensin converting enzyme 2 (ACE2) (Millet, J.K.; Jaimes, A.J.; Whittaker, G.R. Molecular diversity of coronavirus host cell entry receptors. FEMS Microbiol. Rev. 2020, 45, fuaa057.

4. To inhibit DPP4, being a moonlight receptor extensively expressed, must be discussed.

5. Consider the following title:

Unraveling the Interactions between Human DPP4 Receptor and SARS-CoV-2 Variants through Bioinformatics and Molecular Dynamics

MINOR

1. X-ray diffraction data, or X-ray crystallography. Upper X.

2.  Van der Waals' forces are not reported.

3. To add legends to all figures of Supplementary Material

4. Line 144: To replace “…by final short restained molecular…” by “final short restrained molecular…”

5. Lines 145-146: Thus, 200 structures are grouped into 10 clusters.

6. Line 169: “Coulombic” not “columbic”

7. Line 190: The equation must be multiplied by the Planck constant

8. Table 1 is blurred. I suggest writing “Wild type (Wuhan)”

9. Lines 472-475: “We may analyze the effect of clustering of receptors like DPP4 and ACE2 both in association with SARS-CoV-2. From this, we can hypothesize (no predict) the variant associated role in COVID-19 as well as in long COVID-19 due to DPP4 binding.”

I think that this claim can be complemented by the observation that survivors of previous coronavirus infections, including the SARS epidemic of 2003 and the Middle East respiratory syndrome (MERS) outbreak of 2012, have demonstrated a similar constellation of persistent symptoms, reinforcing concern for clinically significant sequelae of COVID-19.

References abound. Examples:

° Ahmed, H. et al. Long-term clinical outcomes in survivors of severe acute respiratory syndrome and Middle East respiratory syndrome coronavirus outbreaks after hospitalisation or ICU admission: a systematic review and meta-analysis. J. Rehabil. Med. 52, jrm00063 (2020).

° Hui, D. S. et al. Impact of severe acute respiratory syndrome (SARS) on pulmonary function, functional capacity, and quality of life in a cohort of survivors. Thorax 60, 401–409 (2005).

° Lam, M. H. et al. Mental morbidities and chronic fatigue in severe acute respiratory syndrome survivors: long-term follow-up. Arch. Intern. Med. 169, 2142–2147 (2009).

There are several rypos. Severl sentences must be improved. Some sentences are unclear.

Author Response

Responses to all the quires of the Reviewer 1 are addressed  and please find the file attached.

Reviewer 2 Report

In this work, the authors performed molecular docking and molecular dynamics simulations to investigate interactions between spike proteins and DPP4. Below are my comments:

1. I suggest the authors briefly introduce HADDOCK and explain why they decided to use it in this work. Showing examples of previous studies using this software with robust results is also helpful to support the results in this work. 

2. Both GROMACS 2018.6 and GROMOS96 are a bit old and out of dated. Why not using a more recent version? Especially because a more recent force field should produce more accurate results.

3. In Figure S1, the y-axis label is distance but it should be RMSD, right? Why the RMSD values are so different between different simulations. Some of them like panel A has a rmsd of 0.5 nm but some have much larger values (panel B has larger than 4 nm rmsd). Please clarify.

4. I suggest the authors add figures to show structures of these studied proteins. Figure 1 provides no meaningful information at all.

5. The authors failed to provide more information of pPerturb server. Any previous studies done using this server obtained reliable predictions? If the authors cannot provide more information to support the reliability of these methods, the results are less convincing in this work.

6. I suggest the authors provide figures showing detected interactions. This will be more straightforward and provide more information than listing them in Table 2.

7. Related to my last point, the authors can perform additional analysis to show the frequency (%) a certain interaction is observed in simulations. This will provide more information than just claiming these interactions exist. Besides, the authors have all necessary data for this analysis. No more simulations are needed.

8. What are different colors in Figure 5? It is not labeled or explained in captions.

9. More introduction of normal mode analysis is needed (similar to my earlier points).

10. The authors mentioned crystal structures are available. I suggest the authors use experimental data to validate these predictions in this work if possible. Then the conclusion will be more valid.

Author Response

Responses to all the quires of the Reviewer 2 are addressed  and please find the file attached.

Round 2

Reviewer 1 Report

Authors answered almost all my concerns.

1. Line 191: coulombic intercations

2. h is the Boltzmann constant. Previously I made a mistake and I wrote Planck constant.

3. Van der Waals' forces are not reported.

4. References 64-66 are not given.

Author Response

We, the authors have addressed point wise responses to the reviewer minor comments and please find the attached pdf file. We thank for their positive comments.

Reviewer 2 Report

The authors addressed all my comments in the revised manuscript. 

Author Response

The authors addressed all my comments in the revised manuscript.

We the authors thank the reviewer for considering our responses in a positive note.